# Polynomial Cost of Adaptation for $\mathcal{X}$-Armed Bandits

**Hédi Hadiji**
Laboratoire de Mathématiques d'Orsay
Université Paris-Sud, Orsay, France
hedi.hadiji@math.u-psud.fr

## Abstract

In the context of stochastic continuum-armed bandits, we present an algorithm that adapts to the unknown smoothness of the objective function. We exhibit and compute a *polynomial cost of adaptation* to the Hölder regularity for regret minimization. To do this, we first reconsider the recent lower bound of Locatelli and Carpentier [21], and define and characterize admissible rate functions. Our new algorithm matches any of these minimal rate functions. We provide a finite-time analysis and a thorough discussion about asymptotic optimality.

## 1 Introduction

Multi-armed bandits are a well-known sequential learning problem. When the number of available decisions is large, some assumptions on the environment have to be made. In a vast line of work (see the literature discussion in Section 1.1), these assumptions show up as nonparametric regularity conditions on the mean-payoff function. If this function is Hölder continuous with constant $L$ and exponent $\alpha$, and if the values of $L$ and $\alpha$ are given to the player, then natural strategies can ensure that the regret is upper bounded by

$$L^{1/(2\alpha+1)}T^{(\alpha+1)/(2\alpha+1)} . \tag{1}$$

Of course, assuming that the player knows $\alpha$ and $L$ is often not realistic. Thus the need for *adaptive* methods, that are agnostic with respect to the true regularity of the mean-payoff function. Unfortunately, Locatelli and Carpentier [21] recently showed that full adaptation is impossible, and that no algorithm can enjoy the same minimax guarantees as when the regularity is given to the player. We persevere and address the question:

*What can the player achieve when the true regularity is completely unknown?*

**A polynomial cost of adaptation** In statistics, minimax adaptation for nonparametric function estimation is a deep and active research domain. In many contexts, sharp adaptation is possible; often, an additional logarithmic factor in the error has to be paid when the regularity is unknown: this is known as the *cost of adaptation*. See e.g., Lepskii [20], Birgé and Massart [4], Massart [22] for adaptive methods, and Cai [9] for a detailed survey of the topic. Under some more exotic assumptions —see e.g., Example 3 of Cai and Low [10] — adapting is significantly harder: there may be a *polynomial cost of adaptation*.

In this paper, we show that in the sequential setting of multi-armed bandits, the necessary exploration forces a similar phenomenon, and we exhibit this polynomial cost of adaptation. To do so, we revisit the lower bounds of Locatelli and Carpentier [21], and design a new algorithm that matches these lower bounds.

As a representative example of our results, our algorithm can achieve, without the knowledge of $\alpha$ and $L$, an unimprovable (up to logarithmic factors) regret bound of order

$$L^{1/(1+\alpha)}T^{(\alpha+2)/(2\alpha+2)} . \tag{2}$$

## 1.1 Related work

**Continuum-armed bandits**    Continuum-armed bandits, with nonparametric regularity assumptions, were introduced by Agrawal [1]. Kleinberg [16] established the minimax rates in the Hölder setting and introduced the CAB1 algorithm. Auer et al. [3] studied the problem with additional regularity assumptions under which the minimax rates are improved. Via different roads, Bubeck et al. [7] and Kleinberg et al. [17] explored further generalizations of these types of regularity, namely the zooming dimension and the near-optimality dimension. Bull [8] exhibited an algorithm that essentially adapts to some cases when the near-optimality dimension is zero.

In all these articles, the mean-payoff function needs to satisfy simultaneously two sets of regularity conditions. The first type is a usual Hölder condition, which ensures that the function does not vary too much around (one of) its maxima. The second type is a "margin condition" that lower bounds the number of very suboptimal arms; in the literature these are defined in many technically different ways. Adapting to the margin conditions is often possible when the Hölder regularity is known. However, all these algorithms require some prior knowledge about the Hölder regularity.

In this paper, we focus on the problem of adapting to Hölder regularity. Accordingly, we call *adaptive* the algorithms that assume no knowledge of the Hölder exponent nor of the Lipschitz constant.

**Adaptation for cumulative regret**    Bubeck et al. [5] introduced the problem of adaptation, and adapted to the Lipschitz constant under extra requirements. An important step was made in Locatelli and Carpentier [21], where it is shown that adaptation at the classical minimax rates is impossible. In the same paper, the authors exhibited some conditions under which full adaptation is achievable, e.g., with knowledge of the value of the maximum, or when the near-optimality dimension is zero.

**Other settings**    For simple regret, the objections against adaptation do not hold, as the objective does not penalize exploration. Adaptation up to polylog factors is done with various (meta-)algorithms. Locatelli and Carpentier [21] sketch out an aggregation approach inspired by Lepski's method, while Valko et al. [24], Grill et al. [14], Shang et al. [23] describe cross-validation methods thanks to which they adapt to the near-optimality dimension with unknown smoothness. As it turns out, this last approach yields clean results with our smoothness assumptions; we write the details in Appendix E.

Recently, Krishnamurthy et al. [18] studied continuum-armed contextual bandits and use a sophisticated aggregation scheme to derive an algorithm that adapts to the Lipschitz constant when $L \geqslant 1$.

## 1.2 Contributions and outline

In this paper, we fully compute the cost of adaptation for bandits with Hölder regularity. In Section 2 we discuss the adaptive (and nonadaptive) lower bounds. We take an asymptotic stance in order to precisely define the objective of adaptation. Doing so, we uncover a family of noncomparable lower bounds for adaptive algorithms (Theorem 1), and define the corresponding notion of optimality: admissibility.

Section 3 contains our main contribution: an admissible adaptive algorithm. We first recall the CAB1 algorithm, which is nonadaptive minimax, and use it as a building block for our new algorithm (Subsection 3.1). This algorithm works in a regime-based fashion. Between successive regimes of doubling lengths, we reset the algorithm and use a new discretization with fewer arms. In order to carry information between the different stages, we use CAB1 in a clever way: besides partitioning the arm space, we add summaries of previous regimes by allowing the algorithm to play according to the empirical distributions of past plays. This is formally described in Subsection 3.2.

A salient difference with all previous approaches is that we zoom out by using fewer and fewer arms. To our knowledge, this is unique, as all other algorithms for bandits zoom in in a way that crucially depends on the regularity parameters. Another important feature of our analysis is that we adapt both to the Hölder exponent $\alpha$ and to the Lipschitz constant $L$. On a technical level, this is thanks to the fact that we do not explicitly choose a grid of regularity parameters, which means that we implicitly handle all values $(L, \alpha)$ simultaneously.

We first give a regret bound in the known horizon case (Subsection 3.2), then we provide an anytime version and we show that they match the lower bounds of adaptation (Subsection 3.3). Finally Section 4 provides the proof of our main regret bound.

## 2   Setup, preliminary discussion

### 2.1   Notation and known results

Let us reintroduce briefly the standard bandit terminology. We consider the arm space $\mathcal{X} = [0,1]$. The environment sets a reward function $f : \mathcal{X} \to [0,1]$. At each time step $t$, the player chooses an arm $X_t \in \mathcal{X}$, and the environment then displays a reward $Y_t$ such that $\mathbb{E}[Y_t \mid X_t] = f(X_t)$, independently from the past. We assume that the variables $Y_t - f(X_t)$ are $(1/4)$-subgaussian conditionnally on $X_t$; this is satisfied if the payoffs are bounded in $[0,1]$ by Hoeffding's lemma.

The objective of the player is to find a strategy that minimizes her *expected cumulative (pseudo-)regret*. If $M(f)$ denotes the maximum value of $f$, the regret at time $T$ is defined as

$$\overline{R}_T = TM(f) - \mathbb{E}\left[\sum_{t=1}^{T} Y_t\right] = TM(f) - \mathbb{E}\left[\sum_{t=1}^{T} f(X_t)\right] . \tag{3}$$

In this paper, we assume that the function $f$ satisfies a Hölder assumption around one of its maxima:

**Definition 1.** *For $\alpha > 0$ and $L > 0$, we denote by $\mathcal{H}(L, \alpha)$ the set of functions that satisfy*

$$\exists\, x^\star \in [0,1] \ \ s.t. \ \ f(x^\star) = M(f) \ \ and \ \ \forall\, x \in [0,1] \quad |f(x^\star) - f(x)| \leqslant L\,|x^\star - x|^\alpha . \tag{4}$$

We are interested in minimax rates of regret when the mean-payoff function $f$ belongs to these Hölder-type classes, i.e., the quantity $\inf\limits_{\text{algorithms}} \ \sup\limits_{f \in \mathcal{H}(L,\alpha)} \ \overline{R}_T$.

**MOSS**   Throughout this paper, we exploit discretization arguments and use a minimax optimal algorithm for finite-armed bandits: MOSS, from Audibert and Bubeck [2]. When run for $T$ rounds on a $K$-armed bandit problem with $(1/4)$-subgaussian noise, and when $T \geqslant K$, its regret is upper-bounded by $18\sqrt{KT}$ (the improved constant is from Garivier et al. [12]).

**Non-adaptive minimax rates**   When the regularity is given to the player, for any $\alpha, L$ and $T$:

$$0.001\, L^{1/(2\alpha+1)} T^{(\alpha+1)/(2\alpha+1)} \leqslant \inf\limits_{\text{algorithms}} \ \sup\limits_{f \in \mathcal{H}(L,\alpha)} \ \overline{R}_T \leqslant 28\, L^{1/(2\alpha+1)} T^{(\alpha+1)/(2\alpha+1)} . \tag{5}$$

This is well-known since Kleinberg [16]. For completeness, we recall how to derive the upper bound in Section 3.1, and the lower bound in Section 2.2.

### 2.2   Lower bounds: adaptation *at usual rates* is not possible

Locatelli and Carpentier [21] prove a version of the following theorem; see our reshuffled and slightly improved proof in Appendix F.

**Theorem** (Variation on Th.3 from [21]). *Let $B > 0$ be a positive number. Let $\alpha, \gamma > 0$ and $L, \ell > 0$ be regularity parameters that satify $\alpha \leqslant \gamma$ and $L \geqslant \ell$.*

*Assume moreover that $2^{-3}\, 12^\alpha B^{-1} \leqslant L \leqslant \ell^{1+\alpha}\, T^{\alpha/2}\, 2^{(1+\alpha)(8-2\gamma)}$. If an algorithm is such that $\sup_{f \in \mathcal{H}(\ell,\gamma)} \overline{R}_T \leqslant B$, then the regret of this algorithm is lower bounded on $\mathcal{H}(L, \alpha)$:*

$$\sup\limits_{f \in \mathcal{H}(L,\alpha)} \overline{R}_T \geqslant 2^{-10}\, T L^{1/(\alpha+1)} B^{-\alpha/(\alpha+1)} . \tag{6}$$

**Remark** (Bibliographical note). *Locatelli and Carpentier [21] consider a more general setting where additional margin conditions are exploited. In our setting, we slightly improve their result by dealing with the dependence on the Lipschitz constant, and by removing a requirement on $B$.*

*In a different context, Krishnamurthy et al. [18] show a variation of this bound where the Lipschitz constant is considered, but only in the case where $\alpha = \gamma = 1$, for $\ell = 1$ and $L \geqslant 1$.*

As explained in Locatelli and Carpentier [21] this forbids adaptation at the usual minimax rates over two regularity classes; we recall how in the paragraph that follows Theorem 1. However this is not the end of the story, as one naturally wonders what is the best the player can do.

To further investigate this question, we discuss it asymptotically by considering the rates at which the minimax regret goes to infinity, therefore focusing on the dependence on $T$. Our main results are completely nonasymptotic, yet we feel the asymptotic analysis of optimality is clearer.

**Definition 2.** *Let $\theta : [0, 1] \to [0, 1]$ denote a nonincreasing function. We say an algorithm achieves adaptive rates $\theta$ if*

$$\forall \varepsilon > 0, \ \forall \alpha, \ L > 0, \quad \limsup_{T \to \infty} \frac{\sup_{f \in \mathcal{H}(L,\alpha)} \overline{R}_T}{T^{\theta(\alpha) + \varepsilon}} < +\infty.$$

We include the $\varepsilon$ in the definition in order to neglect the potential logarithmic factors.

As rate functions are not always comparable for pointwise order, the good notion of optimality is the standard statistical notion of *admissibility* (akin to "Pareto optimality" for game-theorists).

**Definition 3.** *A rate function is said to be* admissible *if it is achieved by some algorithm, and if no other algorithm achieves strictly smaller rates for pointwise order. An algorithm is admissible if it achieves an admissible rate function.*

We recall that a function $\theta'$ is strictly smaller than $\theta$ for pointwise order if $\theta'(\alpha) \leqslant \theta(\alpha)$ for all $\alpha$ and $\theta'(\alpha_0) < \theta(\alpha_0)$ for at least one value of $\alpha_0$.

It turns out we can fully characterize the admissible rate functions by inspecting the lower bounds (6).

**Theorem 1.** *The admissible rate functions are exactly the family*

$$\theta_m : \alpha \mapsto \max\left( m, \ 1 - m \frac{\alpha}{\alpha + 1} \right), \quad m \in [1/2, 1]. \tag{7}$$

This theorem contains two assertions. The lower bound side states that no smaller rate function may be achieved by any algorithm. This side is derived from an asymptotic rewording of lower bound (6), see Proposition 1 stated below (proofs are in Appendix A). The second statement is that the $\theta_m$'s are indeed achieved by an algorithm, which is the subject of Section 3.2.

Figure 1 illustrates how these admissible rates compare to each other, and to the usual minimax rates.

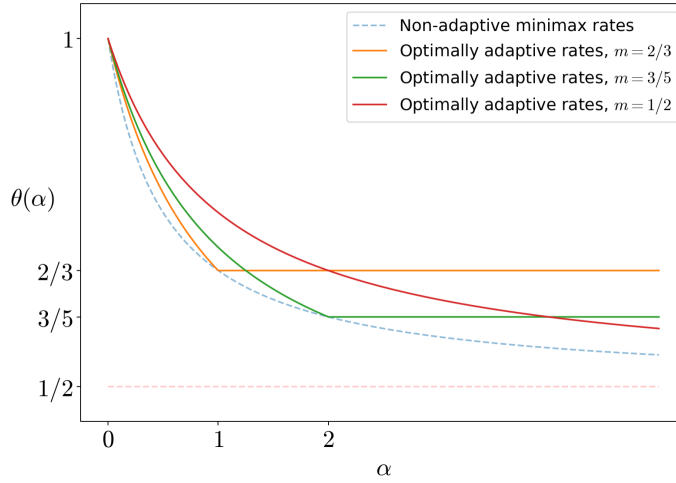

**Figure 1:** The lower bounds on adaptive rates: plots of the admissible rate functions $\alpha \mapsto \theta_m(\alpha)$. If an algorithm has regret of order $\mathcal{O}(T^{\theta(\alpha)})$, then $\theta$ is everywhere above one of these curves.

In particular, we see that reaching the nonadaptive minimax rates for multiple values of $\alpha$ is impossible. Moreover, at $m = (\gamma + 1)/(2\gamma + 1)$, we have $\theta_m(\gamma) = (\gamma + 1)/(2\gamma + 1)$, which is the usual minimax rate (1) when $\gamma$ is known. This yields an alternative parameterization of the family $\theta_m$: one may choose to parameterize the functions either by their value at infinity $m \in [1/2, 1]$, or by the only point $\gamma \in [0, +\infty]$ at which they coincide with the usual minimax rates function (1).

**Proposition 1.** *Assume an algorithm achieves adaptive rates $\theta$. Then $\theta$ satisfies the functional inequation*

$$\forall \gamma > 0, \quad \forall \alpha \leqslant \gamma, \quad \theta(\alpha) \geqslant 1 - \theta(\gamma) \frac{\alpha}{\alpha + 1}. \tag{8}$$

## 2.3 Yet can we adapt in some way?

We have described in (7) the minimal rate functions that are compatible with the lower bounds of adaptation: no algorithm can enjoy uniformly better rates. Of course, at this point, the next natural question is whether any of these adaptive rate functions may indeed be reached by an algorithm.

All previous algorithms for continuum-armed bandits require the regularity as an input in some way (see the literature discussion in Section 1.1). Such algorithms are flawed: if the true regularity is underestimated then we only recover the guarantees that correspond to the smaller regularity, which is often far worse than the lower bounds of Theorem 1. More dramatically, if the true regularity is overestimated, then, a priori, no guarantees hold at all.

We prove that all these rate functions may be achieved by a new algorithm. More precisely, if the player wishes to reach one of the lower bounds $\theta_m$, she may select a value of the input accordingly and match the chosen $\theta_m$. This is our main contribution and is described in the next section.

## 3 An admissible adaptive algorithm and its analysis

We discuss in Section 3.1 how the well-known CAB1 algorithm can be generalized for our purpose. In Section 3.2 we describe our algorithm and the main upper bound on its regret. Section 3.3 is devoted to the anytime version of the algorithm and to a discussion on optimality.

### 3.1 An abstract version of CAB1 as a building block towards adaptation

We describe a generalization of the CAB1 algorithm from Kleinberg [16], where we include arbitrary measures in the discretization. Although this extension is straightforward, we detail it as we will use this algorithm repeatedly further in this paper. In the original CAB1, the space of arms is discretized into a partition of $K$ subsets, and an algorithm for finite-armed bandits plays on the $K$ midpoints of the sets. Auer et al. [3] replace the midpoints by a random point uniformly chosen in the subset.

We introduce a generic version of this algorithm we call CAB1.1. We consider $K$ arbitrary probability distributions over $\mathcal{X}$, which we denote by $(\pi_i)_{1 \leqslant i \leqslant K}$. Denote also by $\pi(f)$ the expectation of $f(X)$ when $X \sim \pi$. At each time step, the decision maker chooses one distribution, $\pi_{I_t}$, and plays an arm picked according to that distribution. By the tower rule, she receives a reward such that

$$\mathbb{E}[Y_t \mid I_t] = \mathbb{E}[f(X_t) \mid I_t] = \pi_{I_t}(f).$$

As the player uses a finite-arm algorithm $\mathcal{A}$ to select $I_t$, the regret she suffers can be decomposed as the sum of two terms (denoting by $\tilde{R}_T$ the expected regret of the finite-armed algorithm):

$$\overline{R}_T = T\big(M(f) - \max_{i=1,\dots,K} \pi_i(f)\big) + \tilde{R}_T\big((\pi_i(f))_{1 \leqslant i \leqslant K}; \mathcal{A}\big). \tag{9}$$

This identity is central to the construction of our algorithm. Using terminology from Auer et al. [3], the first term measures an *approximation error* of the maximum of $f$, and the other the actual *cost of learning* in the approximate problem. Parameters are chosen to balance these two sources of error.

---

**Algorithm 1** CAB1.1 (Continuum-Armed Bandit, adapted from Kleinberg [16])

---

1: **Input**: $T$ the time horizon, $K$ probability measures over $\mathcal{X}$ denoted by $\pi_1, \dots, \pi_K$, discrete $K$-armed bandit algorithm $\mathcal{A}$
2: **for** $t = 1, 2, \dots, T$ **do**
3:     Define $I_t$ the arm in $\{1, \dots, K\}$ recommended by $\mathcal{A}$
4:     Play $X_t \in \mathcal{X}$ drawn according to $\pi_{I_t}$, and receive $Y_t$ such that $\mathbb{E}[Y_t|X_t] = f(X_t)$
5:     Give $Y_t$ as input to $\mathcal{A}$ corresponding to $I_t$
6: **end for**

---

The canonical example is that for which the space of arms is cut into a partition. Denote by $\mathbf{Disc}(K)$ the family of the uniform measures over the intervals $[(i-1)/K, i/K]$ for $1 \leqslant i \leqslant K$. We state this results (and prove it in Appendix A.1) to recall the non-adaptive minimax bound (1).

**Proposition 2.** *Let $\alpha > 0$ and $L > 1/\sqrt{T}$ be regularity parameters, and define the number of discrete arms $K^\star = \min\big(\lceil L^{2/(2\alpha+1)} T^{1/(2\alpha+1)}\rceil, T\big)$. Algorithm CAB1.1 run with the uniform discretization $\mathbf{Disc}(K^\star)$ and $\mathcal{A} = $MOSS enjoys the bound* $\displaystyle\sup_{f \in \mathcal{H}(L,\alpha)} \overline{R}_T \leqslant 28 \, L^{1/(2\alpha+1)} \, T^{(\alpha+1)/(2\alpha+1)}$.

## 3.2 Memorize past plays, Discretize the arm space, and Zoom Out: the MeDZO algorithm

To achieve adaptation, we combine two tricks: *going from fine to coarser discretizations* while *keeping a summary of past plays in memory*.

Our algorithm works in successive regimes. At each time regime $i$, we reset the algorithm and start over a new regime of length double the previous one ($\Delta T_i = 2^{p+i}$), and with fewer discrete arms ($K_i = 2^{p+2-i}$). While doing this, we keep in memory the previous plays: in addition to the uniform distributions over the subsets of partitions, we include the empirical measures $\widehat{\nu}_j$ of the actions played in the past regimes, for $j < i$.

---

**Algorithm 2** MeDZO (Memorize, Discretize, Zoom Out)

---

1: **Input**: parameter $B$, time horizon $T$
2: **Set**: $p = \lceil \log_2 B \rceil$, $K_i = 2^{p+2-i}$ and $\Delta T_i = 2^{p+i}$
3: **for** $i = 1, \ldots, p$ **do**
4:     For $\Delta T_i$ rounds, run algorithm CAB1.1 with the uniform discretization in $K_i$ pieces *and* the empirical measures of the previous plays $\widehat{\nu}_j$ for $j < i$; use MOSS as the discrete algorithm.[a]
5:     **Set**: $\widehat{\nu}_i$ the empirical measure of the plays during regime $i$.
6: **end for**

---

[a]No $\widehat{\nu}$ is used for $i = 0$

Appendix C provides a figure illustrating the behavior of the algorithm.

Our construction is based on the following remark. Consider the approximation error suffered during regime $i$. Denoting the by $\Pi_i$ the set of measures given to the player during regime $i$, that is, the uniform measures over the regular $K_i$-partition and the empirical measures of arms played during the regimes $j < i$, the approximation error is bounded as follows:

$$\Delta T_i \left( M(f) - \mathbb{E}\left[ \max_{\pi \in \Pi_i} \pi(f) \right] \right) \leqslant \Delta T_i \left( M(f) - \mathbb{E}[\widehat{\nu}_j(f)] \right) = \frac{\Delta T_i}{\Delta T_j} \sum_{t \in \text{Regime } j} \left( M(f) - \mathbb{E}[f(X_t)] \right) \quad (10)$$

and this bound is proportional to the regret suffered during regime $j$. This means that even though we zoom out by using fewer arms, we can make sure that the average approximation error in regime $i$ is less than the regret previously suffered. Moreover, the first discretizations are fine enough to ensure a small regret in the first regimes, thanks to the Hölder property. This argument is formalized in the proof (Lemma 1), and shows that MeDZO maintains a balance between approximation and cost of learning that yields optimal regret.

A surprising fact here is that we go from finer to coarser discretizations during the different phases. Thus, paradoxically, *the algorithm zooms out as time passes*. Note also that although this regime-based approach is reminiscent of the doubling trick, there is an essential difference in that information is carried between the regimes via the distribution of the previous plays.

We first state our central result, a generic bound that holds for any input parameter $B$. We discuss the optimality of these adaptive bounds in the next subsection.

**Theorem 2.** *Algorithm 2 run with the knowledge of $T$ and input $B \geqslant \sqrt{T}$ enjoys the following guarantee: for all $\alpha > 0$ and $L > 0$,*

$$\sup_{f \in \mathcal{H}(L, \alpha)} \overline{R}_T \leqslant 412 \left( \log_2 B \right)^{3/2} \max \left( B, \, TL^{1/(\alpha+1)} B^{-\alpha/(\alpha+1)} \right). \quad (11)$$

We provide some illustrative numerical experiments in Appendix D, comparing the results of MeDZO with other non-adaptive algorithms.

## 3.3 Discussion: anytime version and admissibility

**Anytime version via the doubling trick**    The dependence of Algorithm 2 on the parameter $B$ makes it horizon-dependent. We use the doubling trick to build an anytime version of the algorithm. At each new doubling-trick regime, we input a value of $B$ that depends on the length of the $k$-th regime. If it is of length $T^{(k)}$, one typically thinks of $B_k = (T^{(k)})^m$ for some exponent $m$. In that case, we get the following bound —see the proof and description of the algorithm in Appendix B.

**Corollary 1** (Doubling trick version). *Choose $m \in [1/2, 1]$. The doubling-trick version of MeDZO, run with $m$ as sole input (and without the knowledge of T) ensures that for all regularity parameters $\alpha > 0$ and $L > 0$ and for $T \geqslant 1$*

$$\sup_{f \in \mathcal{H}(L, \alpha)} \overline{R}_T \leqslant 4000 (\log_2 T^m)^{3/2} \max\left(T^m, TL^{1/(\alpha+1)}(T^m)^{-\alpha/(\alpha+1)}\right) = \mathcal{O}\left((\log T)^{3/2} T^{\theta_m(\alpha)}\right).$$

**Admissibility of Algorithm 2**    The next result is a direct consequence of Corollary 1. This echoes the discussion following Theorem 1, and shows that for any input parameter $m$, the anytime version of MeDZO cannot be improved uniformly for all $\alpha$.

**Corollary 2.** *For any $m \in [1/2, 1]$, the doubling trick version of MeDZO (see App. B) with input $m$ achieves rate function $\theta_m$, and is therefore admissible.*

## 3.4   About the remaining parameter: the $B = \sqrt{T}$ case

Tuning the value of $B$ amounts to selecting one of the minimal curves in Figure 1. Therefore this parameter is a feature of the algorithm, as it allows the player to choose between the possible optimal behaviors. The tuning of this parameter is an unavoidable choice for the player to make.

The next example illustrates well the performance of MeDZO, as it is easily comparable to the usual minimax bounds. Looking at Figure 1, this choice corresponds to $m = 1/2$, i.e., the only choice of parameter that reaches the usual minimax rates as $\alpha \to \infty$. In other words, if the players wishes to ensure that her regret on very regular functions is of order $\sqrt{T}$, then she has to pay a polynomial cost of adaptation for not knowing $\alpha$ and that price is exactly the ratio between (1) and (2).

**Corollary 3.** *Set a horizon $T$ and run Algorithm 2 with $B = \sqrt{T}$. Then for $\alpha > 0$ and $L > 1/\sqrt{T}$,*

$$\sup_{f \in \mathcal{H}(L, \alpha)} \overline{R}_T \leqslant 146 \left(\log_2 T\right)^{3/2} L^{1/(\alpha+1)} T^{(\alpha+2)/(2\alpha+2)}. \tag{12}$$

This is straightforward from Theorem 2, since the inequality $B = \sqrt{T} \leqslant TL^{1/(\alpha+1)}\sqrt{T}^{-\alpha/(\alpha+1)}$ holds whenever $L \geqslant 1/\sqrt{T}$. An anytime version of this result can be obtained from Corollary 1.

## 4   Proof of Theorem 2

*Full proof of Theorem 2.* Let $\mathcal{F}_t = \sigma(I_1, X_1, Y_1, \dots, I_t, X_t, Y_t)$ be the $\sigma$-algebra corresponding to the information available at the end of round $t$. Define also the transition times $T_i = \sum_{j=1}^{i} \Delta T_j$ with the convention $T_0 = 0$. Let us first verify that $T$ is smaller than the total length of the regimes. By definition of $p$, we have $B \leqslant 2^p < 2B$. Thus $T_p = 2^{p+1}(2^p - 1) \geqslant 2B(B-1) > B^2 > T$, and the algorithm is indeed well-defined up to time $T$.

Consider the regret suffered during the $i$-th regime $\overline{R}_{T_{i-1}, T_i} := \Delta T_i M(f) - \sum_{t=T_{i-1}+1}^{T_i} \mathbb{E}\big[f(X_t)\big]$. We bound this quantity thanks to the decomposition (9), by first conditioning on the past up to time $T_{i-1}$. Since there are $K_i + i$ discrete actions, the regret bound on MOSS ensures that

$$\mathbb{E}\left[\sum_{t=T_{i-1}+1}^{T_i} \left(M(f) - f(X_t)\right) \,\middle|\, \mathcal{F}_{T_{i-1}}\right] \leqslant \Delta T_i \left(M(f) - M_i^\star\right) + 18\sqrt{(K_i + i)\Delta T_i} \tag{13}$$

where $M_i^\star = \max\{\pi_j^{(i)}(f) \mid \pi_j^{(i)} \in \mathbf{Disc}(K_i)\} \cup \big\{\widehat{\nu}_\ell(f) \mid \ell = 0, \dots, i-1\big\}$. Notice that this bound holds even though $M_i^\star$ is a random variable, as the algorithm is completely reset, and the measures $(\widehat{\nu}_j)_{j<i}$ are fixed at time $T_{i-1} + 1$ (i.e., they are $\mathcal{F}_{T_{i-1}}$-measurable). Integrating once more, we obtain

$$\overline{R}_{T_{i-1}, T_i} \leqslant \Delta T_i \left(M(f) - \mathbb{E}[M_i^\star]\right) + 18\sqrt{(K_i + i)\Delta T_i}. \tag{14}$$

**Bounding the cost of learning.**    By definition of $K_i$ and $\Delta T_i$, we have $K_i \Delta T_i = 2^{2p+2} \leqslant 16B^2$. Therefore, since $p$ and $K_i$ are integers greater than 1, using $a + b - 1 \leqslant ab$ for positive integers,

$$\sqrt{(K_i + i)\Delta T_i} \leqslant \sqrt{(K_i + p - 1)\Delta T_i} \leqslant \sqrt{pK_i \Delta T_i} \leqslant 4\sqrt{p}B. \tag{15}$$

**Bounding the approximation error.** The key ingredient for this part is the following fact, that synthetizes the benefits of our construction as hinted in (10) and the surrounding discussion.

**Lemma 1.** *The total approximation error of MeDZO in regime $i$ is controlled by the Hölder bound on the grid of mesh size $1/K_i$, and by the regret suffered during the previous regimes,*

$$\Delta T_i \left( M(f) - \mathbb{E}[M_i^\star] \right) \leqslant \Delta T_i \, \min\left( L \, \frac{1}{K_i^\alpha}, \min_{j<i} \left( \frac{\overline{R}_{T_{j-1},T_j}}{\Delta T_j} \right) \right) \qquad (16)$$

*Proof.* This derives easily from the construction of the algorithm, i.e., from the definition of $M_i^\star$. Considering an interval in the regular $K_i$-partition that contains a maximum of $f$, by the Hölder property, $M(f) - M_i^\star \leqslant L/K_i^\alpha$. For the second minimum, as described in Eq. (10), for $j < i$,

$$M(f) - M_i^\star \leqslant M(f) - \widehat{\nu}_j(f) = \frac{1}{\Delta T_j} \sum_{t=T_{j-1}+1}^{T_j} \left( M(f) - f(X_t) \right).$$

Taking an expectation, $\overline{R}_{T_{j-1},T_j}$ appears, and we conclude by taking the minimum over $j$. $\qquad\square$

Remember that since $K_i \Delta T_i = 2^{2p+2}$, we have $L \Delta T_i / K_i^\alpha = L 2^{2p+2}/K_i^{1+\alpha}$. Therefore, the first bound on the approximation error in (16) increases with $i$, as $K_i$ decreases with $i$. Denote by $i_0$ the last time regime $i$ for which

$$L \frac{\Delta T_{i_0}}{K_{i_0}^\alpha} \leqslant B. \qquad (17)$$

If this is never satisfied, i.e., not even for $i = 1$, then $L2^{p+1}/2^{\alpha(p+1)} > B$ which yields, using $B \leqslant 2^p \leqslant 2B$, that $4LB \geqslant 2^{\alpha+1}B^\alpha B$ and then $L > B^\alpha/2$. In that case, $L^{1/(\alpha+1)}B^{-\alpha/(\alpha+1)} \geqslant 1$ and the total regret bound (11) is true as it is weaker than the trivial bound $R_T \leqslant T$.

Hence we may assume that $i_0 \geqslant 1$ is well defined. By comparing $i$ to $i_0$, we now show the inequality

$$\sum_{i=1}^{p} \Delta T_i \left( M(f) - \mathbb{E}[M_i^\star] \right) \leqslant \sum_{i=1}^{i_0} B + \sum_{i=i_0+1}^{p} 2(1 + 72\sqrt{p})\Delta T_i \, L^{1/(\alpha+1)} B^{-\alpha/(\alpha+1)}. \qquad (18)$$

For all $i \leqslant i_0$ the approximation error is smaller than the first argument of the minimum in (16), and this term is smaller than $B$. Therefore $\Delta T_i \left( M(f) - \mathbb{E}[M_i^\star] \right) \leqslant B$. In particular, this together with (14) and (15) implies that the total regret suffered during regime $i_0$ is $\overline{R}_{T_{i_0-1},T_{i_0}} \leqslant (1 + 72\sqrt{p})B$.

For the later time regimes $i_0 < i \leqslant p$, we use the fact that preceding empirical measures were kept as discrete actions, and in particular the one of the $i_0$-th regime: (16) instantiated with $j = i_0$ yields

$$\Delta T_i \left( M(f) - \mathbb{E}[M_i^\star] \right) \leqslant \Delta T_i \frac{\overline{R}_{T_{i_0-1},T_{i_0}}}{\Delta T_{i_0}} \leqslant \left( 1 + 72\sqrt{p} \right) \Delta T_i \frac{B}{\Delta T_{i_0}}. \qquad (19)$$

Solving equations $L\Delta T_{i_0}/K_{i_0}^\alpha \approx B \approx 4\sqrt{\Delta T_{i_0} K_{i_0}}$, we get $B/\Delta T_{i_0} \leqslant 2\, L^{1/(\alpha+1)}B^{-\alpha/(\alpha+1)}$, (details are in Appendix A.4). Therefore for $i_0 < i \leqslant p$, using (19),

$$\Delta T_i \left( M(f) - \mathbb{E}[M_i^\star] \right) \leqslant 2(1 + 72\sqrt{p}) \, \Delta T_i \, L^{1/(\alpha+1)}B^{-\alpha/(\alpha+1)},$$

and we obtain (18) by summing over $i$.

**Conclusion** We conclude with some crude boundings. First, as $i_0 \leqslant p$ and the sum of the $\Delta T_i$'s is smaller than $T$, the total approximation error is less than $pB + 2(1+72\sqrt{p})TL^{1/(\alpha+1)}B^{-\alpha/(\alpha+1)}$. Let us include the cost of learning, which is smaller than $72p\sqrt{p}B$ and conclude, using $a+b \leqslant \max(a,b)$

$$\begin{aligned}
\overline{R}_T &\leqslant 2(1 + 72\sqrt{p})TL^{1/(\alpha+1)}B^{-\alpha/(\alpha+1)} + pB + 72p^{3/2}B \\
&= 2(1 + 72\sqrt{p})TL^{1/(\alpha+1)}B^{-\alpha/(\alpha+1)} + p(1 + 72\sqrt{p})B \\
&\leqslant \left( 2(1 + 72\sqrt{p}) + p(1 + 72\sqrt{p}) \right) \max\left( B, \, TL^{1/(\alpha+1)}B^{-\alpha/(\alpha+1)} \right)
\end{aligned} \qquad (20)$$

from which the desired bound follows, using $1 \leqslant p$, and $p \leqslant 2\log_2 B$ and $4(1 + 72\sqrt{2}) \leqslant 412$. $\quad\square$

# 5  Further considerations

**Local regularity assumption**   Theorem 2 holds under a relaxed smoothness assumption, namely that the function satisfies the Hölder condition only in a small cell containing the maximum. By looking carefully at the proof, we observe that the condition is only required up to the $i_0$-th epoch (defined in (17)), at which the size of the cells in the discretization is of order $1/K_{i_0} \approx (LB)^{-1/(1+\alpha)}$. Therefore we only need condition (4) to be satisfied for points $x$ in an interval of size $(LB)^{-1/(1+\alpha)}$ around the maximum.

**Higher dimension**   Our results can be generalized to functions $[0,1]^d \to [0,1]$ that are $\|\cdot\|_\infty$-Hölder. For MeDZO to be well-defined, take $K_i = 2^{d(p+2-i)}$ and $\Delta T_i = 2^{d(p+i)}$, with $p \approx (\log B)/d$. The bounds are similar to their one-dimensional counterparts, up to replacing $\alpha$ by $\alpha/d$ in the exponents, but the constants are deteriorated by a factor that is exponential in $d$. The bound in Theorem 2 changes to $\max\left(B, L^{d/(\alpha+d)} T B^{-\alpha/(\alpha+d)}\right)$.

## Acknowledgements

We would like to thank Gilles Stoltz and Pascal Massart for their valuable comments and suggestions.

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
