[Supplementary Material]

# Supplementary Material for
# "Polynomial Cost of Adaptation for $\mathcal{X}$-Armed Bandits"

## A   Omitted proofs

### A.1   Proposition 2: Regret bound for non-adaptive CAB1.1

This proof is a straightforward application of the Hölder bound and of the bound of MOSS, together with the approximation/cost of learning decomposition of the regret. Some extra care is needed to handle the boundary cases.

*Proof of Proposition 2.* Choose $f \in \mathcal{H}(L, \alpha)$. Let us denote by $i^\star$ an integer such that there exists an optimal arm $x^\star$ in the interval $\left[(i^\star - 1)/K^\star, i^\star/K^\star\right]$. By the Hölder assumption

$$\frac{1}{K^\star} \int_{(i^\star-1)/K^\star}^{i^\star/K^\star} \left(f(x^\star) - f(x)\right) \mathrm{d}x \leqslant L \left(\frac{1}{K^\star}\right)^\alpha,$$

and this upper bounds the approximation error of the discretization. Moreover, since $T \geqslant K^\star$, the cost of learning is smaller than $18\sqrt{K^\star T}$. Thus by (9)

$$\overline{R}_T \leqslant TL \left(\frac{1}{K^\star}\right)^\alpha + 18\sqrt{K^\star T}.$$

$K^\star$ was chosen to minimize this quantity. We distinguish cases depending on the value of $K^\star$.

If $1 < K^\star < T$, then $L^{2/(2\alpha+1)}T^{1/(2\alpha+1)} \leqslant K^\star \leqslant 2L^{2/(2\alpha+1)}T^{1/(2\alpha+1)}$ (the bound $\lceil x \rceil \leqslant 2x$, which is valid when $x \geqslant 1$, is more practical to handle the multiplicative constants), we deduce the upper bound:

$$\left(1 + 18\sqrt{2}\right) L^{1/(2\alpha+1)}T^{(\alpha+1)/(2\alpha+1)}.$$

Since we assumed that $L > 1/\sqrt{T}$, we have always $K^\star > 1$. Therefore the last case to consider is if $K^\star = T$. Then $L^{2/(2\alpha+1)}T^{1/(2\alpha+1)} \geqslant T/2$ and thus $L \geqslant 2^{-(2\alpha+1)/2} T^\alpha$. In this case $L^{1/(2\alpha+1)}T^{(\alpha+1)/(2\alpha+1)} \geqslant (\sqrt{2}/2)T$ and the claimed bound is met since in that case, we have by a trivial bound $\overline{R}_T \leqslant T \leqslant \sqrt{2}\, L^{1/(2\alpha+1)}T^{(\alpha+1)/(2\alpha+1)}$. $\qquad\square$

### A.2   Proposition 1: Lower bound on the adaptive rates

*Proof of Proposition 1.* Choose $\alpha, \gamma$ such that $\alpha \leqslant \gamma$, and $\varepsilon > 0$. Set $L > 0$. There exist constants $c_1$ and $c_2$ (depending on $L, \alpha, \gamma$ and $\varepsilon$) such that for $T$ large enough,

$$\sup_{f \in \mathcal{H}(L,\alpha)} \overline{R}_T \leqslant c_1 T^{\theta(\alpha)+\varepsilon} \quad \text{and} \quad \sup_{f \in \mathcal{H}(L,\gamma)} \overline{R}_T \leqslant c_2 T^{\theta(\gamma)+\varepsilon}.$$

Moreover, for $T$ large enough, the assumptions for lower bound (6) hold. Hence applying the lower bound with $B = c_2 T^{\theta(\gamma)+\varepsilon}$, for some constant $c_3$:

$$c_1 T^{\theta(\alpha)+\varepsilon} \geqslant 0.0001\, T \left(c_2 T^{\theta(\gamma)+\varepsilon}\right)^{-\alpha/(\alpha+1)} \geqslant c_3\, T^{1-\theta(\gamma)\alpha/(\alpha+1)-\varepsilon\alpha/(\alpha+1)}$$

Since the above inequality holds for any $T$ sufficiently large, this implies that for all $\varepsilon > 0$

$$\theta(\alpha) + \varepsilon \geqslant 1 - \theta(\gamma)\frac{\alpha}{\alpha+1} - \varepsilon\frac{\alpha}{\alpha+1},$$

which yields the desired result as $\varepsilon \to 0$. $\qquad\square$

### A.3 Theorem 1: Admissible rate functions

We prove here that all the admissible rate functions belong to the family $(\theta_m)$, by relying on Proposition 1. The proof is done through a careful inspection of the functional inequation defining the lower bound.

*Proof of Theorem 1.* First of all, by Corollary 2, the appropriately tuned MeDZO may achieve all the $\theta'_m s$. Thus we are left to prove the lower bound side, i.e., that all the admissible rate functions belong to the family $\theta_m$.

The best way to see this is to first notice that for $\theta$ nonincreasing and positive, the inequation in Proposition 1 is equivalent to

$$\forall \alpha > 0, \quad \theta(\alpha) \geqslant 1 - \theta(\infty)\frac{\alpha}{\alpha + 1}. \tag{21}$$

Notice that taking $\gamma = +\infty$ is always valid in what follows, as $\theta$ is assumed to be nonincreasing and lower bounded by $1/2$. Now if $\theta$ satisfies (8), then it satisfies (21) by taking $\gamma = +\infty$. For the converse, consider $\alpha \leqslant \gamma$, then $\theta(\gamma) \geqslant \theta(\infty)$, thus $1 - \theta(\infty)\alpha/(\alpha + 1) \geqslant 1 - \theta(\gamma)\alpha/(\alpha + 1)$.

Now consider an admissible $\theta$. Since $\theta$ is achieved by some algorithm, by Proposition 1 and the remark above, it satisfies Eq. (21). As $\theta$ is nonincreasing, and by Eq. (21), we have $\theta(\alpha) \geqslant \theta(\infty)$ and $\theta(\alpha) \geqslant 1 - \theta(\infty)\alpha/(\alpha + 1)$. In other words, $\theta \geqslant \theta_{m_\theta}$, where $m_\theta = \theta(\infty) \in [1/2, 1]$. By the admissibility of $\theta$, this implies that $\theta = \theta_{m_\theta}$. $\qquad\square$

### A.4 Calculations in the proof of Theorem 2

*Details on* (18), *in the proof of Theorem 2.* By definition of $i_0$, and since we assumed that $i_0 < p$

$$B \leqslant L \frac{\Delta T_{i_0+1}}{K_{i_0+1}^\alpha},$$

i.e., using $K_{i_0} \Delta T_{i_0} = 2^{2p+2}$,

$$B \leqslant 2^{1+\alpha} L \frac{\Delta T_{i_0}}{K_{i_0}^\alpha} = 2^{1+\alpha} L \left(\Delta T_{i_0}\right)^{1+\alpha} 2^{-(2p+2)\alpha}.$$

From this we deduce, using $2^p \geqslant B$ for the second inequality,

$$\left(\Delta T_{i_0}\right)^{(1+\alpha)} \geqslant 2^{-1-\alpha} B L^{-1} 2^{(2p+2)\alpha} \geqslant 2^{-1+\alpha} L^{-1} B^{2\alpha+1}.$$

Hence, using $2^{(\alpha-1)/(\alpha+1)} \geqslant 1/2$, we obtain $\Delta T_{i_0} \geqslant (1/2) L^{-1/(\alpha+1)} B^{(2\alpha+1)/(\alpha+1)}$, thus $B/\Delta T_{i_0} \leqslant 2 L^{1/(\alpha+1)} B^{-\alpha/(\alpha+1)}$. $\qquad\square$

## B Anytime-MeDZO and proof

The doubling trick is the most standard way of converting non-anytime algorithms into anytime algorithms, when the regret bound is polynomial. It consists in taking fresh starts of the algorithm over a grid of dyadic times. The implementation of the trick is straightforward in our case.

---

**Algorithm 3** Doubling trick MeDZO

---

1: **Input**: parameter $m \in [1/2, 1]$;
2: **for** $i = 0, \ldots$ **do**
3:     Run MeDZO (Alg. 2) with input $B = 2^{im}$ for $2^i$ rounds
4: **end for**

---

**Corollary** (Doubling trick version). *Choose $m \in [1/2, 1]$. The doubling-trick version of MeDZO, run with $m$ as sole input (and without the knowledge of T) ensures that for all regularity parameters $\alpha > 0$ and $L > 0$ and for $T \geqslant 1$*

$$\sup_{f \in \mathcal{H}(L,\alpha)} \overline{R}_T \leqslant 4000(\log_2 T^m)^{3/2} \max\left(T^m, T L^{1/(\alpha+1)}(T^m)^{-\alpha/(\alpha+1)}\right) = \mathcal{O}\left((\log T)^{3/2} T^{\theta_m(\alpha)}\right).$$

As the regret bound is not exactly of the form $cT^\theta$, we work with the polynomial version of the bound on the regret of MeDZO, equation (20), for the doubling trick to be effective. Obviously the value of the constant in the bound is not our main focus, but we still write it explicitly as it shows that there is no hidden dependence on the various parameters.

*Proof.* By (20), with $p_i = \lceil \log_2 2^{im} \rceil \leqslant 1 + \log_2 2^{im}$, in the $i$-th doubling trick regime, the cumulative regret is bounded by

$$2(1 + 72\sqrt{1 + \log_2 2^{im}})2^i L^{1/(\alpha+1)}(2^{im})^{-\alpha/(\alpha+1)} + (1 + \log_2 2^{im})\big(1 + 72\sqrt{1 + \log_2 2^{im}}\big)2^{im}$$

Now since

$$\sum_{i=0}^{\lceil \log_2 T \rceil} 2^i = 2^{\lceil \log_2 T \rceil + 1} - 1 \geqslant 2T - 1 \geqslant T\,,$$

there are always less than $\lceil \log_2 T \rceil$ full regimes. Therefore, using $\log_2 2^{im} \leqslant \log_2 T^m$, and summing over the regimes, the first part of this sum is bounded by

$$2(1 + 72\sqrt{2\log_2 T^m})L^{1/(\alpha+1)} \sum_{i=0}^{\lceil \log_2 T \rceil} 2^{i(1 - m\alpha/(\alpha+1))}$$

$$\leqslant 2(1 + 72\sqrt{2\log_2 T^m})L^{1/(\alpha+1)} \frac{2^{(\lceil \log_2 T \rceil + 1)(1 - m\alpha/(\alpha+1))}}{2^{1 - m\alpha/(\alpha+1)} - 1}$$

$$\leqslant 2(1 + 72\sqrt{2})\sqrt{\log_2 T^m}L^{1/(\alpha+1)} \frac{2^{2(1 - m\alpha/(\alpha+1))}}{\sqrt{2} - 1}T(T^m)^{-\alpha/(\alpha+1)}$$

$$\leqslant 2(1 + 72\sqrt{2})\sqrt{\log_2 T^m}L^{1/(\alpha+1)} \frac{4}{\sqrt{2} - 1}T(T^m)^{-\alpha/(\alpha+1)}$$

where we used $2^{\lceil \log_2 T \rceil} \leqslant 2T$; we also used the fact that since $m \geqslant 1/2$, we always have the inequality $1 - m\alpha/(\alpha + 1) \geqslant 1/2$ to bound the denominator. Similarly, the second part is bounded by

$$2(1 + 72\sqrt{2})(\log_2 T^m)^{3/2} \sum_{i=0}^{\lceil \log_2 T \rceil} 2^{im} \leqslant 2(1 + 72\sqrt{2})(\log_2 T^m)^{3/2}\frac{4}{\sqrt{2} - 1}T^m\,.$$

All in all, we obtain the same minimax guarantees as if we had known the time horizon in advance, but with an extra multiplicative factor of $4/(\sqrt{2} - 1) \approx 9,66$. □

## C  Illustration

In this section we provide a figure to illustrate the behavior of MeDZO in a schematic example.

MeDZO starts by playing on a fine discretization with a size of order $\sqrt{T}$, but for a short length of time, of order $\sqrt{T}$. At the end of the first epoch, it memorizes the empirical distribution of the arms played; then it runs a new instance of CAB1.1 with both the coarser discretization, and the memorized action. This process is repeated until the time horizon is reached.

The payoffs of the memorized actions increase until the size of the discretization reaches a critical value; after that they fluctuate. Therefore MeDZO manages to maintain a regret of order the approximation error at this critical discretization, multiplied by $T$.

**Figure 2:** Behavior of MeDZO on a schematic drawing. The expected payoffs of the memorized actions are displayed in red; those from the usual discretization are in blue.

## D Numerical experiments

This section contains some numerical experiments comparing the regrets of algorithms that require the knowledge of the smoothness, against MeDZO.

We examine bandit problems defined by their mean-payoff functions and gaussian $\mathcal{N}(0; 1/4)$ noise. The functions considered are $f : x \mapsto (1/2)\sin(13x)\sin(27x) + 0.5$ taken from Bubeck et al. [7], $g : x \mapsto \max\left(3.6\,x(1-x), 1 - 1/0.05\,|x - 0.05|\right)$ adapted from Coquelin and Munos [11] and the Garland function $x \mapsto x(1-x)(4 - \sqrt{|\sin(60x)|}$, which we took from Valko et al. [24]. The functions are plotted in Figure 3.

**(a)** $f$       **(b)** $g$       **(c)** The Garland function

**Figure 3:** Problems considered

The algorithms we compare are SR from Locatelli and Carpentier [21], and CAB1 from Kleinberg [16] with MOSS as the discrete algorithm. SR takes directly the smoothness $\alpha$ as an input, and assumes $L = 1$. For CAB1, we compute the optimal discretization size for $L = 1$ and varying $\alpha$.

**(a)** $f$  **(b)** $g$  **(c)** The Garland function

**Figure 4:** Regrets of MeDZO, and of SR and CAB1 run with different values of the smoothness parameter.

In Figure 4 we plot the cumulative regret of the algorithms after a time horizon $T = 300000$, for varying values of the assumed smoothness. For each problem, MeDZO was run only once, as it does not need to know the smoothness. The regret was averaged over $N = 75$ runs, and the dotted curves represent +/- one standard deviation.

We recall that minimax guarantees are worst-case guarantees, therefore comparing algorithms on a single problem can only serve as an empirical illustration.

As expected, the regrets of both SR and CAB1 depend on some careful tuning of the input parameter, determined by the smoothness. The optimal tuning is unclear, and seems to vary on the algorithm. MeDZO, on the other hand, obtains reasonable regret with no tuning. Surprisingly, CAB1 with overestimated smoothness seems to behave quite well, although the large variance sometimes makes it difficult to distinguish the results. Recall that MeDZO is the only algorithm with theoretical guarantees for high values of $\alpha$.

## E   About simple regret

In this section, we consider the case of *simple regret*, which complements the discussion about adaptation to smoothness in sequential optimization procedures. We write out how to achieve adaptation at usual rates for simple regret under Hölder smoothness assumptions. We do not claim novelty here, as adaptive strategies have already been used for simple regret under more sophisticated regularity conditions (see, e.g., Grill et al. [14], Shang et al. [23] and a sketched out procedure in Locatelli and Carpentier [21]); however, we feel the details deserve to be written out in this simpler setting.

Let us recall the definition of simple regret. In some cases, we may only require that the algorithm outputs a recommendation $\tilde{X}_T$ at the end of the $T$ rounds, with the aim of minimizing the simple regret, defined as

$$\overline{r}_T = M(f) - \mathbb{E}\big[f\big(\tilde{X}_T\big)\big] \,.$$

This setting is known under various names, e.g., pure exploration, global optimization or black-box optimization. As noted in Bubeck et al. [6], minimizing the simple regret is easier than minimizing the cumulative regret in the sense that if the decision-maker chooses a recommendation uniformly among the arms played $X_1, \dots, X_T$, then

$$\overline{r}_T = M(f) - \frac{1}{T} \sum_{t=1}^{T} \mathbb{E}\big[f(X_t)\big] = \frac{\overline{R}_T}{T} \,. \tag{22}$$

The minimax rates of simple regret over Hölder classes $\mathcal{H}(L, \alpha)$ are lower bounded by $\Omega(L^{1/(2\alpha+1)} T^{-\alpha/(2\alpha+1)})$, which are exactly the rates for cumulative regret divided by $T$ (see Locatelli and Carpentier [21] for a proof of the lower bound). Consequently, at known regularity, any minimax optimal algorithm for cumulative regret automatically yields a minimax recommendation for simple regret via (22).

When the smoothness is unknown, the situation turns out to be quite different. Adapting to the Hölder parameters can be done at only a (poly-)logarithmic cost for simple regret, contrasting with the polynomial cost of adaptation of cumulative regret. This can be achieved thanks to a very general and simple cross-validation scheme defined in Shang et al. [23], named General Parallel Optimization.

**Algorithm 4** GPO (General Parallel Optimization) for Hölder minimax adaptation
___

1: **Input**: time horizon $T \geqslant 8$
2: **Set**: $p = \lceil \log_2 T \rceil$ and define $K_i = 2^i$ for $i = 1, \ldots, p$
3: **for** $i = 1, \ldots, p$ **do**                                                              // Exploration
4:     For $\lfloor T/(2p) \rfloor$ rounds, run algorithm CAB1.1 with the discretization in $K_i$ pieces; use MOSS as the discrete algorithm
5:     Define output recommendation $\tilde{X}^{(i)}$, uniformly chosen among the $\lfloor T/(2p) \rfloor$ arms played
6: **end for**
7: **for** $i = 1, \ldots, p$ **do**                                                              // Cross-validation
8:     Play $\lfloor T/(2p) \rfloor$ times each recommendation $\tilde{X}^{(i)}$ and compute the average reward $\hat{\mu}^{(i)}$
9: **end for**
10: **return** A recommendation $\tilde{X}_T = \tilde{X}^{(\hat{\imath})}$ with $\hat{\imath} \in \arg\max \hat{\mu}^{(i)}$
___

The next result shows that the player obtains the same simple regret bounds as when the smoothness is known (up to logarithmic factors).

**Theorem 3.** *GPO with CAB1.1 as a sub-algorithm (Alg. 4) achieves, given $T \geqslant 8$ and without the knowledge of $\alpha$ and $L$, for all $\alpha > 0$ and $L \geqslant 2^{\alpha+1/2}\sqrt{\lceil \log_2 T \rceil / T}$ the bound*

$$\sup_{f \in \mathcal{H}(L,\alpha)} \overline{r}_T \leqslant \left(54 + \frac{\sqrt{\pi}}{2}\log_2 T\right)L^{1/(2\alpha+1)}\left(\frac{\lceil \log_2 T \rceil}{T}\right)^{\alpha/(2\alpha+1)} = \tilde{\mathcal{O}}\left(L^{1/(2\alpha+1)}T^{-\alpha/(2\alpha+1)}\right).$$

The $\tilde{\mathcal{O}}$ notation hides the $\log T$ factors, and the assumption that $T \geqslant 8$ is needed to ensure that $T/(2p) = T/(2\lceil \log_2 T \rceil) \geqslant 1$: otherwise the algorithm itself is ill-defined.

*Proof.* Let $f \in \mathcal{H}(L, \alpha)$ denote a mean-payoff function. Once again we decompose the error of the algorithm into two sources. The simple regret is the sum of the regret of the best recommendation among the $p$ received, $r_{\min}$, and of a cross-validation error, $r_{\mathrm{CV}}$,

$$M(f) - \mathbb{E}[f(\tilde{X}_T)] = \underbrace{\min_{i=1,\ldots,p}\left(M(f) - \mathbb{E}\left[f(\tilde{X}^{(i)})\right]\right)}_{r_{\min}} + \underbrace{\max_{i=1,\ldots,p}\left(\mathbb{E}\left[f(\tilde{X}^{(i)})\right] - \mathbb{E}\left[f(\tilde{X}_T)\right]\right)}_{r_{\mathrm{CV}}}. \tag{23}$$

We now show that $r_{\mathrm{CV}} \leqslant p^{3/2}\sqrt{\pi/(4T)}$, by detailing an argument that is sketched in the proof of Thm. 3 in Shang et al. [23]. Denote by $\hat{\mu}^{(i)}$ the empirical reward associated to recommendation $i$, and $\hat{\imath} = \arg\max \hat{\mu}^{(i)}$, so that $\tilde{X}_T = \tilde{X}^{(\hat{\imath})}$. Then for any fixed $i$, by the tower rule,

$$\mathbb{E}\left[\hat{\mu}^{(i)}\right] = \mathbb{E}\left[\mathbb{E}\left[\hat{\mu}^{(i)} \,\middle|\, \tilde{X}^{(i)}\right]\right] = \mathbb{E}\left[f(\tilde{X}^{(i)})\right]. \tag{24}$$

Therefore, by the above remarks, and since $\hat{\mu}^{(i)} \leqslant \hat{\mu}^{(\hat{\imath})}$,

$$\mathbb{E}\left[f(\tilde{X}^{(i)})\right] - \mathbb{E}\left[f(\tilde{X}_T)\right] = \mathbb{E}\left[\hat{\mu}^{(i)} - f(\tilde{X}^{(\hat{\imath})})\right] \leqslant \mathbb{E}\left[\hat{\mu}^{(\hat{\imath})} - f(\tilde{X}^{(\hat{\imath})})\right].$$

We have to be careful here, as $\hat{\imath}$ is a random index that depends on the random variables $\hat{\mu}^{(i)}$'s: we cannot apply directly the tower rule as in (24). To deal with this, let us use an integrated union bound. Denote by $(\cdot)^+$ the positive part function, then

$$\mathbb{E}\left[\hat{\mu}^{(\hat{\imath})} - f(\tilde{X}^{(\hat{\imath})})\right] \leqslant \mathbb{E}\left[\left(\hat{\mu}^{(\hat{\imath})} - f(\tilde{X}^{(\hat{\imath})})\right)^+\right] \leqslant \sum_{j=1}^{p}\mathbb{E}\left[\left(\hat{\mu}^{(j)} - f(\tilde{X}^{(j)})\right)^+\right],$$

and we are back to handling empirical means of i.i.d. random variables. For each $j$, the reward given $\tilde{X}^{(j)}$ is $(1/4)$-subgaussian. Therefore, as $\hat{\mu}^{(i)}$ is the empirical mean of $n = \lfloor T/(2p) \rfloor$ plays of the same arm $\tilde{X}^{(j)}$, this mean $\hat{\mu}^{(i)}$ is $(1/(4n))$-subgaussian conditionally on $\tilde{X}^{(j)}$ and thus for all $\varepsilon > 0$

$$\mathbb{P}\left[\hat{\mu}^{(j)} - f(\tilde{X}^{(j)}) \geqslant \varepsilon\right] \leqslant e^{-2n\varepsilon^2}.$$

Hence by integrating over $\varepsilon \in [0, +\infty)$, using Fubini's theorem, a change of variable $x = \sqrt{4n\varepsilon}$ (and using the fact that $\lfloor T/(2p) \rfloor \geqslant T/(4p)$ as $T/(2p) \geqslant 1$):

$$
\mathbb{E}\left[\left(\hat{\mu}^{(j)} - f(\tilde{X}^{(j)})\right)^+\right] = \int_0^{+\infty} \mathbb{P}\left[\hat{\mu}^{(j)} - f(\tilde{X}^{(j)}) \geqslant \varepsilon\right] \mathrm{d}\varepsilon
$$

$$
\leqslant \int_0^{+\infty} e^{-2n\varepsilon^2} \, \mathrm{d}\varepsilon = \frac{1}{\sqrt{4n}} \int_0^{+\infty} e^{-x^2/2} \, \mathrm{d}x
$$

$$
= \sqrt{\frac{\pi}{8n}} = \sqrt{\frac{\pi}{8 \lfloor T/2p \rfloor}} \leqslant \sqrt{\frac{\pi p}{4T}}
$$

Putting back the pieces together, we have shown that for any $i$,

$$
\mathbb{E}\left[f(\tilde{X}^{(i)})\right] - \mathbb{E}\left[f(\tilde{X}_T)\right] \leqslant \sum_{j=1}^p \sqrt{\frac{\pi p}{4T}} = p^{3/2} \sqrt{\frac{\pi}{4T}}.
$$

We deduce the same bound for $r_{\mathrm{CV}}$ by taking the maximum over $i$.

Let us now bound $r_{\min}$. By Eq. (9), using the fact that $\lfloor T/(2p) \rfloor \geqslant T/(4p)$ as $T/(2p) \geqslant 1$, for all $i$

$$
M(f) - \mathbb{E}\left[f(\tilde{X}^{(i)})\right] \leqslant \frac{L}{K_i^\alpha} + 18\sqrt{\frac{4pK_i}{T}}.
$$

We summarize a few calculations in the next lemma. These calculations come from the minimization over the $K_i$'s of the previous bound, with a case disjunction arising from the boundary cases.

**Lemma 2.** *At least one of the three following inequalities holds :*

$$
L < 2^{\alpha+1/2} \sqrt{\frac{p}{T}} \quad \text{or} \quad L \geqslant T^\alpha \sqrt{p}
$$

*or*

$$
\min_{i=1,\ldots,p} \left(\frac{L}{K_i^\alpha} + 36\sqrt{\frac{pK_i}{T}}\right) \leqslant 53 L^{1/(2\alpha+1)} \left(\frac{p}{T}\right)^{\alpha/(2\alpha+1)}.
$$

Let us consider these three cases separately. The first one is forbidden by the assumption that $L \geqslant 2^{\alpha+1/2} \sqrt{p/T}$. In the second case, the function is so irregular that the claimed bound becomes worse than $\overline{r}_T \leqslant 56 \, p^{1/2+\alpha/(2\alpha+1)}$, which is weaker than the trivial bound $\overline{r}_T \leqslant 1$.

Finally, in the third case, we may assume that $L \geqslant 2^{\alpha+1/2} \sqrt{p/T} \geqslant \sqrt{p/T}$. Then we have

$$
L^{1/(2\alpha+1)} \geqslant \left(\frac{p}{T}\right)^{1/(2(2\alpha+1))} = \left(\frac{p}{T}\right)^{1/2} \left(\frac{p}{T}\right)^{-\alpha/(2\alpha+1)},
$$

and thus $\sqrt{p/T} \leqslant L^{1/(2\alpha+1)}(p/T)^{\alpha/(2\alpha+1)}$. By injecting the bound of Lemma 2 and the bound on $r_{\mathrm{CV}}$ into (23):

$$
\overline{r}_T \leqslant 53 L^{1/(2\alpha+1)} \left(\frac{p}{T}\right)^{\alpha/(2\alpha+1)} + p\sqrt{\frac{\pi}{4}}\sqrt{\frac{p}{T}} \leqslant (53 + p\sqrt{\pi/4}) L^{1/(2\alpha+1)} \left(\frac{p}{T}\right)^{\alpha/(2\alpha+1)}
$$

and the stated bound holds, since $53 + p\sqrt{\pi/4} \leqslant 53 + (\log_2 T + 1)\sqrt{\pi/4} \leqslant 54 + \sqrt{\pi/4}\log_2 T$. $\square$

*Proof of Lemma 2.* We upper bound the minimum by comparing the two quantities

$$
\frac{L}{K_i^\alpha} \quad \text{v.s.} \quad \sqrt{\frac{pK_i}{T}}.
$$

As the first term is decreasing with $i$, and the second term is increasing with $i$, two extreme cases have te be dealt with. If the first term is always smaller than the second, i.e., even for $i = 1$, then:

$$
\frac{L}{2^\alpha} < \sqrt{\frac{p\,2}{T}}.
$$

This is the first case in the statement of the lemma. Otherwise, the first term might always be greater than the second one, i.e., even for $i = p$ and

$$\frac{L}{2^{\alpha p}} \geqslant \sqrt{\frac{p2^p}{T}}$$

which is equivalent to

$$L^2 \geqslant p\frac{2^{p(2\alpha+1)}}{T} \, ,$$

hence, since $2^p \geqslant T$,

$$L^2 \geqslant pT^{2\alpha}$$

which is exactly the second inequality of our statement.

Otherwise, define $i^\star$ to be an index such that

$$\frac{L}{K_{i^\star-1}^\alpha} \geqslant \sqrt{\frac{pK_{i^\star-1}}{T}} \quad \text{and} \quad \frac{L}{K_{i^\star}^\alpha} \leqslant \sqrt{\frac{pK_{i^\star}}{T}} \tag{25}$$

By the preceding discussion, $i^\star$ is well defined and $1 < i^\star \leqslant p$. Then by definition of $i^\star$ (the first equation in (25))

$$2^{\alpha+1/2}\frac{L}{K_{i^\star}^\alpha} \geqslant \sqrt{\frac{pK_{i^\star}}{T}} \, .$$

Hence, by squaring and regrouping the terms

$$K_{i^\star}^{2\alpha+1} \leqslant 2^{2\alpha+1}L^2\frac{T}{p}$$

thus

$$K_{i^\star} \leqslant 2L^{2/(\alpha+1)}\left(\frac{T}{p}\right)^{1/(2\alpha+1)}$$

and

$$\sqrt{\frac{pK_{i^\star}}{T}} \leqslant \sqrt{2}L^{1/(2\alpha+1)}\left(\frac{p}{T}\right)^{\alpha/(2\alpha+1)}$$

and finally, recalling the second equation in (25)

$$\frac{L}{K_{i^\star}^\alpha} + 36\sqrt{\frac{pK_{i^\star}}{T}} \leqslant 37\sqrt{\frac{pK_{i^\star}}{T}} \leqslant 37\sqrt{2}L^{1/(2\alpha+1)}\left(\frac{p}{T}\right)^{\alpha/(2\alpha+1)} \, .$$

$$\square$$

## F  Proof of our version of the lower bound of adaptation

Here we provide the full proof of our version of the lower bound of adaptation stated in Section 2.2.

Our statement differs from that of Locatelli and Carpentier [21] on some aspects. First, and most importantly, we include the dependence on the Lipschitz constants, and we do not consider margin regularity. We also remove a superfluous requirement on $B$, that $B \leqslant c\,T^{(\alpha+1)/(2\alpha+1)}$, which was just an artifact of the original proof. Furthermore we believe that the additional condition that $L \leqslant \mathcal{O}(T^{\alpha/2})$ in our version was implicitly used in this original proof. Finally, the value of the constant differs, partly because of the analysis, and partly because we consider $(1/4)$-subgaussian noise instead of 1-subgaussian noise.

We managed to obtain these improvements thanks to a different proof technique. In the original proof, the authors compare the empirical likelihoods of different outcomes and use the Bretagnolle-Huber inequality. We choose to build the lower bound in a slightly different way (see Garivier et al. [13]): we handle the changes of measure implicitly thanks to Pinsker's inequality (Lemma 3). Following Lattimore and Szepesvári [19], we also chose to be very precise in the definition of the bandit model, in order to make rigorous a few arguments that are often used implicitly in the literature on continuous bandits.

The main argument of the proof, that is, the sets of functions considered, are already present in Locatelli and Carpentier [21].

Before we start with the proof, let us state a technical tool. Denote by KL the Kullback-Leibler divergence. The next lemma is a generalized version of Pinsker's inequality, tailored to our needs.

**Lemma 3.** *Let $\mathbb{P}$ and $\mathbb{Q}$ be two probability measures. For any random variable $Z \in [0,1]$,*

$$|\mathbb{E}_{\mathbb{P}}[Z] - \mathbb{E}_{\mathbb{Q}}[Z]| \leqslant \sqrt{\frac{\mathrm{KL}(\mathbb{P}, \mathbb{Q})}{2}}$$

*Proof.* For $z \in [0,1]$, by the classical version of Pinsker's inequality applied to the event $\{Z \geqslant z\}$:

$$|\mathbb{P}[Z \geqslant z] - \mathbb{Q}[Z \geqslant z]| \leqslant \sqrt{\frac{\mathrm{KL}(\mathbb{P}, \mathbb{Q})}{2}}\,.$$

Therefore, by Fubini's theorem and the triangle inequality, and by integrating the preceding inequality:

$$|\mathbb{E}_{\mathbb{P}}[Z] - \mathbb{E}_{\mathbb{Q}}[Z]| = \left| \int_0^1 \big(\mathbb{P}[Z \geqslant z] - \mathbb{Q}[Z \geqslant z]\big)\mathrm{d}z \right| \leqslant \int_0^1 |\mathbb{P}[Z \geqslant z] - \mathbb{Q}[Z \geqslant z]|\mathrm{d}z \leqslant \sqrt{\frac{\mathrm{KL}(\mathbb{P}, \mathbb{Q})}{2}}$$

$\square$

*Proof of the lower bound.* For the sake of completeness, we recall in detail the construction of Locatelli and Carpentier [21], with some minor simplifications that fit our setting. Fix regularity parameters $\ell, L, \alpha$ and $\gamma$ satisfying $\ell \leqslant L$ and $\gamma \geqslant \alpha$, so that $\mathcal{H}(\ell, \gamma) \subset \mathcal{H}(L, \alpha)$ (remember the functions are defined on $\mathcal{X} = [0, 1]$).

Fix $M \in [1/2, 1]$. Let $K \in \mathbb{N} \setminus \{0\}$ and $\Delta \in \mathbb{R}_+$ be some parameters of the construction whose values will be determined by the analysis. We define furthermore a partition of $[0, 1]$ into $K+1$ sets, $H_0 = [1/2, 1]$ and $H_i = [(i-1)/(2K), i/(2K)]$ for $1 \leqslant i \leqslant K$, along with their middle points $x_i \in H_i$. Finally, define the set of hypotheses $\phi_i$ for $i = 0, \dots, K$ as follows

$$\phi_i(x) = \begin{cases} \max\big(M - \Delta,\ M - \Delta/2 - \ell\,|x - x_0|^\gamma\big) & \text{if } x \in H_0\,, \\ \max\big(M - \Delta,\ M - L\,|x - x_i|^\alpha\big) & \text{if } x \in H_i \text{ and } s \neq 0\,, \\ M - \Delta & \text{otherwise.} \end{cases} \tag{26}$$

**Figure 5:** Mean-payoff functions for the lower bound

Figure 5 illustrates how the $\phi_i$'s are defined : for $1 \leqslant i \leqslant K$, the function $\phi_i$ displays a peak of size $\Delta$ and of low regularity $(L, \alpha)$, localized in $H_i$, and another peak of size $\Delta/2$, of higher

regularity $(\ell, \gamma)$ in $H_0$. The function $\phi_0$ only has the peak of size $\Delta/2$ and regularity $(\ell, \gamma)$. We need to add requirements on the values of the parameters, to make sure the indeed functions belong to the appropriate regularity classes. These requirements are written in the following lemma, which we prove later.

**Lemma 4.** *If* $(\Delta/L)^{1/\alpha} \leqslant 1/(4K)$ *then* $\phi_0 \in \mathcal{H}(\ell, \gamma)$, *and if* $(\Delta/(2\ell))^{1/\gamma} \leqslant 1/4$ *then* $\phi_i \in \mathcal{H}(L, \alpha)$ *for* $i \geqslant 1$.

Fix a given algorithm. The idea of the proof of the lower bound is to use the fact that if the player has low regret, that is, less than $B$, when the mean-payoff function is $\phi_0 \in \mathcal{H}(L, \alpha)$, then she has to play in $H_0$ often. This in turn constrains the amount of exploration she can afford, and limits her ability to find the maximum when the mean-payoff functions is $\phi_i$ for $i > 0$.

**Canonical bandit model**   In this paragraph, we build the necessary setting for a rigorous development. The continous action space gives rise to measurability issues, and one should be particularly careful when handling changes of measure as we do here. Following Lattimore and Szepesvári [19, Chap. 4.7, 14 (Ex.11) and 15 (Ex.8) ], we build the canonical bandit model in order to apply the chain rule for Kullback-Leibler divergences rigorously. To our knowledge, this is seldom done carefully, the two notable exceptions being the above reference and Garivier et al. [13]. We also use the notion of probability kernels in this paragraph; see Kallenberg [15, Chap. 1 and 5] for a definition and properties.

Define a sequence of measurable spaces $\Omega_t = \prod_{s=1}^{t} \mathcal{X} \times \mathbb{R}$, together with their Borel $\sigma$-algebra (with the usual topology on $\mathcal{X} = [0, 1]$ and on $\mathbb{R}$). We call $h_t = (x_1, y_1, \ldots, x_t, y_t) \in \Omega_t$ a history up to time $t$. By an abuse of notation, we consider that $\Omega_t \subset \Omega_{t'}$ when $t \leqslant t'$.

An algorithm is a sequence $(K_t)_{1 \leqslant t \leqslant T}$ of (regular) probability kernels, with $K_t$ from $\Omega_{t-1}$ to $\mathcal{X}$, modelling the choice of the arm at time $t$. By an abuse of notation, the first kernel $K_1$ is an arbitrary measure on $\mathcal{X}$, the law of the first arm picked. Define for each $i$ another probability kernel modelling the reward obtained: $L_{i,t}$ from $\Omega_t \times \mathcal{X}$ to $\mathbb{R}$. We write it explicitly as :

$$L_{i,t}\big((x_1, y_1, \ldots, x_t), B\big) = \sqrt{\frac{2}{\pi}} \int_B e^{-2\big(x - \phi_i(x_t)\big)^2} \, \mathrm{d}x$$

These kernels define probability laws $\mathbb{P}_{i,t} = L_{i,t}(K_t \mathbb{P}_{i,t-1})$ over $\Omega_t$. Doing so, we ensured that under $\mathbb{P}_{i,t}$ the coordinate random variables $X_t : \Omega_t \to \mathcal{X}$ and $Y_t : \Omega_t \to \mathbb{R}$), defined as $X_t(x_1, \ldots, x_t, y_t) = x_t$ and $Y_t(x_1, \ldots, x_t, y_t) = y_t$ are such that given $X_t$, the reward $Y_t$ is distributed according to $\mathcal{N}\big(\phi_i(X_t), 1/4\big)$. Denote by $\mathbb{E}_i$ the expectation taken according to $\mathbb{P}_{i,t}$. We also index recall the pseudo-regret: $\overline{R}_{T,i} = TM(\phi_i) - \mathbb{E}_i\Big[\sum_{t=1}^{T} \phi_i(X_t)\Big]$.

A rewriting of the chain rule for Kullback-Leibler divergence with our notation would be (see Lattimore and Szepesvári [19, Exercise 11 Chap. 14] for a proof)

**Proposition** (Chain rule). *Let* $\Omega$ *and* $\Omega'$ *be measurable subsets of* $\mathbb{R}^d$ *equipped with their natural* $\sigma$-*algebra. Let* $\mathbb{P}$ *and* $\mathbb{Q}$ *be probability distributions defined over* $\Omega$, *and* $K$ *and* $L$ *be regular probability kernels from* $\Omega$ *to* $\Omega'$ *then*

$$\mathrm{KL}\big(K\mathbb{P}, L\mathbb{Q}\big) = \mathrm{KL}(\mathbb{P}, \mathbb{Q}) + \int_\Omega \mathrm{KL}\big(K(\omega, \cdot), L(\omega, \cdot)\big) \, d\mathbb{P}(\omega)$$

The key assumptions are that $\Omega$ and $\Omega'$ are subspaces of $\mathbb{R}^d$, and that $K$ and $L$ satisfy measurability conditions, as they are regular kernels; these assumptions justify the heavy setting we introduced.

Under this setting, we may call to the chain rule twice to see that for any $t$:

$$
\begin{aligned}
\mathrm{KL}\left(\mathbb{P}_0^t, \mathbb{P}_i^t\right) &= \mathrm{KL}\left(L_{0,t}(K_t\mathbb{P}_0^{t-1}), L_{i,t}(K_t\mathbb{P}_i^{t-1})\right) \\
&= \mathrm{KL}\left(K_t\mathbb{P}_0^{t-1}, K_t\mathbb{P}_i^{t-1}\right) + \int_{\Omega_{t-1}\times\mathcal{X}} \mathrm{KL}\left(L_{0,t}(h_{t-1}, x_t, \cdot), L_{i,t}(h_{t-1}, x_t, \cdot)\right) \mathrm{d}K_t\mathbb{P}_0^{t-1}(h_{t-1}, x_t) \\
&= \mathrm{KL}\left(\mathbb{P}_0^{t-1}, \mathbb{P}_i^{t-1}\right) + \int_{\Omega_{t-1}\times\mathcal{X}} \mathrm{KL}\left(L_{0,t}(h_{t-1}, x_t, \cdot), L_{i,t}(h_{t-1}, x_t, \cdot)\right) \mathrm{d}K_t\mathbb{P}_0^{t-1}(h_{t-1}, x_t) \\
&= \mathrm{KL}\left(\mathbb{P}_0^{t-1}, \mathbb{P}_i^{t-1}\right) + \int_{\mathcal{X}} \mathrm{KL}\left(\mathcal{N}(\phi_0(x_t), 1/4), \mathcal{N}(\phi_i(x_t), 1/4)\right) \mathrm{d}\mathbb{P}_0^{t-1}(x_t) \\
&= \mathrm{KL}\left(\mathbb{P}_0^{t-1}, \mathbb{P}_i^{t-1}\right) + \mathbb{E}_0\left[\mathrm{KL}\left(\mathcal{N}(\phi_0(X_t), 1/4), \mathcal{N}(\phi_i(X_t), 1/4)\right)\right]
\end{aligned}
$$

where the penultimate equality comes from the fact that the density of the kernel $L_{i,t-1}$ depends only on the last coordinate $x_t$, and is exactly that of a gaussian variable.

We obtain the KL decomposition by iterating $T$ times,

$$
\mathrm{KL}\left(\mathbb{P}_0^T, \mathbb{P}_i^T\right) = \mathbb{E}_0\left[\sum_{t=1}^{T} \mathrm{KL}\left(\mathcal{N}(\phi_0(X_t), 1/4), \mathcal{N}(\phi_i(X_t), 1/4)\right)\right]
$$

**Continuation of the proof**    Let us also define $N_{H_i}(T) = \sum_{t=1}^{T} \mathbb{1}_{\{X_t \in H_i\}}$ the number of times the algorithm selects an arm in $H_i$. The hypotheses $\phi_i$ were defined for the three following inequalities to hold. For all $i \geqslant 1$:

$$
\overline{R}_{T,i} \geqslant \frac{\Delta}{2}\left(T - \mathbb{E}_i\left[N_{H_i}(T)\right]\right) = \frac{T\Delta}{2}\left(1 - \frac{\mathbb{E}_i\left[N_{H_i}(T)\right]}{T}\right), \tag{27}
$$

$$
\overline{R}_{T,0} \geqslant \frac{\Delta}{2}\sum_{i=1}^{K} \mathbb{E}_0\left[N_{H_i}(T)\right], \tag{28}
$$

and

$$
\begin{aligned}
\mathrm{KL}(\mathbb{P}_0^T, \mathbb{P}_i^T) &= \mathbb{E}_0\left[\sum_{t=1}^{T} \mathrm{KL}\left(\mathcal{N}(\phi_0(X_t), 1/4), \mathcal{N}(\phi_i(X_t), 1/4)\right)\right] \\
&= \mathbb{E}_0\left[\sum_{t=1}^{T} 2\left(\phi_0(X_t) - \phi_i(X_t)\right)^2\right] \leqslant 2\,\mathbb{E}_0\left[N_{H_i}(T)\right]\Delta^2.
\end{aligned} \tag{29}
$$

The first two inequalities come from the fact that, under $\mathbb{P}_i$, the player incurs an instantaneous regret of less than $\Delta/2$ whenever she picks an arm outside the optimal cell $H_i$. For the third inequality, first apply the chain rule to compute the Kullback-Leibler divergence, then the inequality is a consequence of the fact that $\phi_i$ and $\phi_0$ differ only in $H_i$, and their difference is less than $\Delta$.

We may now proceed with the calculations. By Lemma 3 applied to the random variable $N_{H_i}(T)/T$:

$$
\frac{\mathbb{E}_i\left[N_{H_i}(T)\right]}{T} \leqslant \frac{\mathbb{E}_0\left[N_{H_i}(T)\right]}{T} + \sqrt{\frac{\mathrm{KL}(\mathbb{P}_0^T, \mathbb{P}_i^T)}{2}}. \tag{30}
$$

We will now show that

$$
\frac{1}{K}\sum_{i=1}^{K} \overline{R}_{T,i} \geqslant \frac{T\Delta}{2}\left(1 - \frac{1}{K} - \sqrt{\frac{\Delta\,\overline{R}_{T,0}}{K}}\right). \tag{31}
$$

Indeed by (in order) averaging (27) over $i = 1, \ldots, K$, using (30), the concavity of $\sqrt{\cdot}$ and (29)

$$\frac{1}{K} \sum_{i=1}^{K} \overline{R}_{T,i} \geqslant \frac{T\Delta}{2} \left( 1 - \frac{1}{K} \sum_{i=1}^{K} \frac{\mathbb{E}_i \left[ N_{H_i}(T) \right]}{T} \right)$$

$$\geqslant \frac{T\Delta}{2} \left( 1 - \frac{1}{K} \sum_{i=1}^{K} \frac{\mathbb{E}_0 \left[ N_{H_i}(T) \right]}{T} - \frac{1}{K} \sum_{i=1}^{K} \sqrt{\frac{\mathrm{KL}(\mathbb{P}_0^T, \mathbb{P}_i^T)}{2}} \right)$$

$$\geqslant \frac{T\Delta}{2} \left( 1 - \frac{1}{K} - \sqrt{\frac{1}{2K} \sum_{i=1}^{K} \mathrm{KL}(\mathbb{P}_0^T, \mathbb{P}_i^T)} \right)$$

$$\geqslant \frac{T\Delta}{2} \left( 1 - \frac{1}{K} - \sqrt{\frac{\Delta^2}{K} \sum_{i=1}^{K} \mathbb{E}_0 \left[ N_{H_i}(T) \right]} \right) .$$

This yields the claimed inequality (31) thanks to (28).

Let us assume for now that $K \geqslant 2$ and $\phi_0 \in \mathcal{H}(\ell, \gamma)$. Then by the assumption on the algorithm, $\overline{R}_{T,0} \leqslant B$, and therefore

$$\frac{1}{K} \sum_{i=1}^{K} \overline{R}_{T,i} \geqslant \frac{T\Delta}{2} \left( \frac{1}{2} - \sqrt{\frac{\Delta B}{K}} \right) . \tag{32}$$

To optimize this bound, we take $\Delta$ as large as possible, while still ensuring that $\sqrt{\Delta B / K}$ is small enough, e.g., less than $1/4$. Furthermore, we impose that the $\phi_i$'s belong to $\mathcal{H}(L, \alpha)$, i.e., by Lemma 4, that $(\Delta/L)^{1/\alpha} \leqslant 1/(4K)$. This leads to the choice

$$\Delta = c \, L^{1/(\alpha+1)} B^{-\alpha/(\alpha+1)} \quad \text{and} \quad K = \left\lfloor \frac{1}{4} \left( \frac{\Delta}{L} \right)^{-1/\alpha} \right\rfloor = \left\lfloor \frac{c^{-1/\alpha}}{4} (LB)^{1/(\alpha+1)} \right\rfloor ,$$

with $c = 1/128$.

**Conclusion, assuming that $K \geqslant 2$ and $\phi_0 \in \mathcal{H}(\ell, \gamma)$** With this choice of parameters, we have by definition of $\Delta$,

$$\Delta B = c \, (LB)^{1/(\alpha+1)} ,$$

and by definition of $K$, since $K \geqslant (c^{-1/\alpha}/8)(LB)^{1/(\alpha+1)}$,

$$\frac{\Delta B}{K} \leqslant 8 c^{1+1/\alpha}$$

hence, using $c^{1/(2\alpha)} \leqslant 1$

$$\sqrt{\frac{\Delta B}{K}} \leqslant 2\sqrt{2} c^{1/2+1/(2\alpha)} \leqslant 2\sqrt{2} \cdot 2^{-7/2} = \frac{1}{4} .$$

With this in hand, we may now go back to inequality (32) to see that

$$\frac{1}{K} \sum_{i=1}^{K} \overline{R}_{T,i} \geqslant \frac{T\Delta}{2} \left( \frac{1}{2} - \frac{1}{4} \right) \geqslant \frac{T\Delta}{8} = \frac{c}{8} \, TL^{1/(\alpha+1)} B^{-\alpha/(\alpha+1)} .$$

By the defintion of $K$, it is always true that $(\Delta/L)^{1/\alpha} \leqslant 1/(4K)$, and therefore, by Lemma 4, all the $\phi_i$'s automatically belong to $\mathcal{H}(L, \alpha)$. Therefore, for all $i$, we have $\sup_{f \in \mathcal{H}(L, \alpha)} \overline{R}_T \geqslant \overline{R}_{T,i}$. Hence, recalling that $c = 1/128$,

$$\sup_{f \in \mathcal{H}(L, \alpha)} \overline{R}_T \geqslant \frac{1}{K} \sum_{i=1}^{K} \overline{R}_{T,i} \geqslant 2^{-10} \, TL^{1/(\alpha+1)} B^{-\alpha/(\alpha+1)} .$$

**Regularity conditions on the mean-payoff functions** $\phi_i$   We now check that $K \geqslant 2$, and that $\phi_0 \in \mathcal{H}(\ell, \gamma)$. Let us first focus on $\phi_0$. By Lemma 4, it is enough to impose that $(\Delta/(2\ell))^{1/\gamma} \leqslant 1/4$, i.e., that

$$c\, L^{1/(\alpha+1)} B^{-\alpha/(\alpha+1)}/(2\ell) \leqslant (1/4)^\gamma$$

that is,

$$L^{1/(\alpha+1)} B^{-\alpha/(\alpha+1)} \leqslant 2\ell(1/4)^\gamma/c = \ell\, 2^{1-2\gamma} c^{-1}\,,$$

i.e., when

$$LB^{-\alpha} \leqslant \ell^{1+\alpha}\, 2^{(1-2\gamma)(1+\alpha)} c^{-(1+\alpha)}$$

hence, replacing $c$ by its value $c = 2^{-7}$, the next condition is sufficient to ensure the regularity of the hypothesis:

$$L \leqslant \ell^{1+\alpha}\, B^\alpha\, c^{-(1+\alpha)}\, 2^{(1+\alpha)(1-2\gamma)} = \ell^{1+\alpha}\, B^\alpha\, 2^{(1+\alpha)(8-2\gamma)}\,,$$

which is one of the two conditions in the statement of the theorem. For the bound to be valid, we must also make sure that $K \geqslant 2$:

$$\left\lfloor \left( \frac{c^{-1/\alpha}}{4} (LB)^{1/(\alpha+1)} \right) \right\rfloor \geqslant 2\,.$$

This condition is weaker than

$$\frac{c^{-1/\alpha}}{4} (LB)^{1/(\alpha+1)} \geqslant 3$$

which is equivalent to

$$L \geqslant c^{(\alpha+1)/\alpha}\, 12^{\alpha+1} B^{-1} = 2^{-7} \cdot 12 \cdot 2^{-6/\alpha} 12^\alpha B^{-1}\,.$$

To ensure this, we require the stronger (but more readable) condition that $L \geqslant 2^{-3} 12^\alpha B^{-1}$.   □

*Proof of Lemma 4.* A good look at Figure 6 should convince the reader of the statement. We wish to make sure that the functions $\phi_i$'s satisfy (4), a Hölder condition around their maximum (and only around this maximum). Given the definition of the functions $\phi_i$, we simply have to check that there is no discontinuity at the boundary of the cell $H_i$. We write out the details for $i > 0$ to remove any doubt; the same analysis can be carried to check that $\phi_0 \in \mathcal{H}(\ell, \gamma)$.

**(a)** $(\Delta/L)^{1/\alpha} \leqslant 1/(4K)$ hence $\phi_i \in \mathcal{H}(L, \alpha)$     **(b)** $(\Delta/L)^{1/\alpha} > 1/(4K)$ hence $\phi_i \notin \mathcal{H}(L, \alpha)$

**Figure 6:** $\phi_i$ is in $\mathcal{H}(L, \alpha)$ if it is everywhere above the green dotted curve $x \mapsto M - L\,|x - x_i|^\alpha$, that is, if the cell $H_i$ has enough room to contain the whole peak of size $\Delta$

For $i > 0$, the function $\phi_i$ reaches its maximum at $x_i = (i-1/2)/2K$, and the value of the maximum is $M$. Then for $x \in H_i$, by definition of $\phi_i$:

$$\phi_i(x) = \max\left( M - \Delta, M - L\,|x_i - x|^\alpha \right) \geqslant M - L\,|x_i - x|^\alpha$$

thus

$$\phi_i(x_i) - \phi_i(x) = M - \phi_i(x) \leqslant L\,|x_i - x|^\alpha\,,$$

Now consider $x \notin H_i$. Assume, as in the statement of the lemma, that $1/(4K) \geqslant (\Delta/L)^{1/\alpha}$. If $x$ is outside of $H_i$, then since $H_i$ is of half-width $1/4K$,

$$|x_i - x| \geqslant \frac{1}{4K} \geqslant \left(\frac{\Delta}{L}\right)^{1/\alpha} \tag{33}$$

and, by definition of $\phi_i$, for all $x$ (even for $x \in H_0$), $\phi_i(x) \geqslant M - \Delta$. Therefore, by (33),

$$\phi_i(x_i) - \phi_i(x) \leqslant \Delta \leqslant L\,|x_i - x|^{\alpha}\ .$$

For all values of $x$, the Hölder condition is satisfied and $\phi_i \in \mathcal{H}(L, \alpha)$.

For $\phi_0$, the same calculations show that there is no jump at the boundary of $[1/2, 1]$, of half-width $1/4$, when the peak is of height $\Delta/2$ and regularity $(\ell, \gamma)$ if $\left((\Delta/2)/\ell\right)^{1/\gamma} \leqslant 1/4$. $\qquad\square$