[Reviews · NeurIPS 2019]

Reviewer 1



The current paper is mathematically well written, but lacks explanations. The results are established using a compilation of techniques from several articles (Kleinberg [15], Audibert and Bubbeck [2], and Garivier et al [11]). However, even if it seems sound, and can be followed line by line, it sometimes lacks explanations about global intuitions. For example, I believe that the proof of Lemma 1 is not mandatory (and its statement is anyway very technical), and it would be better to focus on global explanations of the methodology. As an example, Algorithm 2 (MeDZO) is helpful to discretize the continuous interval: it would be good to explain why this algorithm? Why this selection of the parameters p, K_i and Delta T_i is done in that way works? The horizon T is assumed to be a prior knowledge. This should be stated and commented (if this is indeed the case). In the current form, the proofs are postponed to very technical statements at the end, but the article would really benefit about shorter sketch proofs, with key ideas and explanations. Why this algorithm, and not another one? The current audience of this article, in its current form is for a specialized audience, but not for a large audience corresponding to NIPS.

Reviewer 2



Originality: The results of this paper include some clever use of existing results and techniques along with some novel algorithmic elements. Quality: The technical results of the paper are stated precisely and are accompanied with rigorous proofs. The formal statements are also complemented by the descriptions of their underlying intuition, which helps in understanding the results. Clarity: The paper is very well written, and while the technical content is dense, it was reasonably easy to follow. Significance: I feel that the algorithmic ideas presented for achieving the optimal adaptive rates are important, and can have potential applications in related problems. ------------------------- AFTER AUTHOR RESPONSE: I have read the author response, and I thank the authors for their clarifications. I will retain my previous score for this submission.

Reviewer 3



This paper considers the regret of X-armed bandit problem. They propose a lower bound on the regret of this problem. Then they show their MeDZO algorithm, which has the advantage that need nothing to be the input, and can achieve good performances for any fixed parameters. They also show the theoratical regret upper bound of this algorithm. The idea of using the empirical distribution before is very interesting, which can reduce the number of arms we needed while keeps the regret in a low level. The regret upper bound of this policy matches with the lower bound provided in this paper, which means both bounds are tight. I think I have read some papers that use a similar idea, but this one is the first one I know about using this trick on X-armed bandit model. The proof seems to be correct, but I do not check thoses ones in supplementary file in detail. The writting is also clear for me to understand this paper. I found there are no experiments in this paper, and I am wondering how much the regret gap will be when comparing the MeDZO policy with other ones who takes \alpha as input and then make the optimization. After rebuttal: - I saw the experimental results, and I think add some experiments into the paper is useful for readers to understand the contribution of your papers. - The similar ideas: for example, in "Reducing Dueling Bandits to Cardinal Bandits", their Doubler and MultiSBM policy both regard an empirical distribution on arms as a generalized arm.

[Author Response · NeurIPS 2019]

# Author feedback for
# "Polynomial Cost of Adaptation $\mathcal{X}$-Armed Bandits"

We thank the reviewers for their overall positive and constructive comments.

To answer some concerns of Reviewer #1, we would like to reemphasize the significance of the paper. Adapting to the unknown smoothness in X-armed bandits has been an open problem since Bubeck, Munos, Stoltz, and Szepesvári [3], and a few partial answers have been published since then (described the literature review). This paper completes the picture in the minimax Hölder setting.

Moreover, the Hölder assumption (stated under various names) is standard in this line of work, e.g., in Agrawal [1], Kleinberg [4], Auer et al. [2], Bubeck et al. [3] and Locatelli and Carpentier [6]. Hölder regularity is also omnipresent in non-parametric statistics.

Furthermore, as stressed by Reviewer #2, the paper introduces novel algorithmic ideas that could be applied to other settings. The vanilla X-armed bandits model can be extended in many ways, the first one coming to mind being the contextual setting of Krishnamurthy et al. [5].

Therefore, we believe a strong point of the paper lies in the algorithmic idea together with the proof techniques, and this is why we insisted on being complete and thorough in the mathematical steps.

That said, we acknowledge the technicality of the paper. As suggested by Reviewer # 2, we will add an illustrative figure describing how the algorithm behaves and giving some intuition. We will also follow the recommendation of Reviewer #3 and add the following numerical experiments, for illustrative purposes. We recall that it is quite standard (albeit unfortunate), that papers in this area do not include experiments.

**(a)** CAB1                    **(b)** Subroutine

**Figure 1:** Average regrets of CAB1 and Subroutine (from Locatelli and Carpentier [6]) tuned with varying values of $\alpha$, and of Medzo, after $T = 300000$ time steps. The algorithms were run 30 times and the error bars are 1.96 times the standard deviation. The problem used is $x \mapsto (1/2)\sin(13x)\sin(27x) + 0.5$, taken from Valko et al. [7]

With no knowledge of the true regularity, Medzo obtains a regret that is almost the same as that of algorithms optimally tuned. Intriguingly, CAB1 performs quite well when the smoothness is overestimated, although the variance becomes quite high. Experiments will be further commented in the final version.

Specific points :

R1: "The horizon T is assumed to be a prior knowledge. This should be stated and commented [...]. " : Indeed, Subsection 3.3 and Appendix B discuss this and describe how we can get rid this requirement. In the final version we will recall that by "anytime" we mean without the knowledge of T.

R2 "Can this algorithmic technique deal with cases in which the function is *spatially inhomogenous*, for instance if the Hölder exponent $\alpha$ varies with the input point x." : This is a good point. Our guarantees hold if the Hölder property is satisfied in a small neighbourhood around the maximum, but the minimal size of this neighbourhood depends on T. Our analysis only requires that in every discretizations, (i.e., in the first), the average payoff of the cell containing the optimal action is close to optimal. This is actually the case in previous papers (Zooming, HOO, Locatelli and Carpentier). We will add this remark in the main paper, with some detail in the appendix.

R3 : "I think I have read some papers that use a similar idea, but this one is the first one I know about using this trick on X-armed bandit model." : We would be happy to read (and cite) these works if you find them.



[Meta-Review · NeurIPS 2019]

The reviewers are in agreement that the paper makes good progress on adaptivity of x-armed bandit problems, specifically providing an agnostic algorithm that achieves Pareto optimal regret rates, which matches regret lower bounds for adaptive algorithms. The algorithm that achieves this goal is also of independent interest.